# A Panel of Circulating Non-Coding RNAs in the Diagnosis and Monitoring of Therapy in Egyptian Patients with Breast Cancer

**DOI:** 10.3390/biomedicines11020563

**Published:** 2023-02-15

**Authors:** Nadine Wehida, Wafaa Abdel-Rehim, Hazem El Mansy, Ahmed Karmouty, Maher A. Kamel

**Affiliations:** 1Department of Biomolecular Sciences, Kingston University London, Kingston-upon-Thames KT1 2EE, UK; 2Department of Biochemistry, Medical Research Institute, Alexandria University, Alexandria 21561, Egypt; 3Department of Cancer Management and Reseach, Medical Research Institute, Alexandria University, Alexandria 21561, Egypt; 4Department of Experimental and Clinical Surgery, Medical Research Institute, Alexandria University, Alexandria 21561, Egypt

**Keywords:** breast, cancer, non-coding, RNA, diagnostic, H19, microRNA-675, microRNA let 7, CA 15-3

## Abstract

**Background:** Non-coding RNAs (ncRNAs) have recently been identified to have a pivotal role in many diseases, including breast cancer (BC). This study aims to investigate the relative quantification of long non-coding RNA (lncRNA) H19, microRNA (miR) 675-5p, 675-3p, and miR-let 7 in breast cancer patients. **Methods:** The study was performed on three groups: Group 1: 30 non-intervened BC female patients about to undergo breast surgery; group 2: 30 postoperative female BC patients about to receive adjuvant anthracycline chemotherapy; and group 3: 30 apparently healthy female volunteers as the control group. Plasma samples were drawn before and after the intervention in groups 1 and 2, with a single sample drawn from group 3. The relative quantification levels were compared with healthy control subjects and were related with the clinicopathological statuses of these patients. **Results:** There was a statistically significant increase in H19, miR-675-5p, miR-675-3p, and miR-let 7 in the non-intervened BC patients when compared to the control group. Surgery resulted in a significant reduction in all four ncRNAs under investigation. Chemotherapy brought about a significant increase in the level of miR-let 7, with no significant effect on the remaining parameters measured. The assay discriminated normal from BC where a receiver operating characteristic for the area under the curve (ROC_AUC_) of miR-675-3p showed the maximal AUC of 1.000. The diagnostic sensitivity and specificity were also 100% when CA 15-3 and H19 were combined. **Conclusion**: The results strongly indicate that the panel of ncRNAs in this study can all potentially act as novel biomarkers whether alone or combined in the diagnosis of BC.

## 1. Introduction

NcRNAs constitute 99% of the gross cellular RNA [1] and are classified broadly into housekeeping and regulatory ncRNAs. Ribosomal RNA (rRNA), transfer RNA (tRNA), small nuclear RNA (snRNA), and small nucleolar RNA (snoRNA) constitute the housekeeping RNAs. Regulatory RNAs are further classified into long non coding RNAs (>200 nucleotides) and short ncRNAs (<200 nucleotides) [2].

Findings of research studies on the diagnostic power of lncRNA are still conflicting and controversial. LncRNA H19, a member on the lncRNA family, is a paternally imprinting gene located in chromosome 11p15.5. It has recently been linked with various human cancers [3]. It has been suggested that lncRNA H19 has a role as an oncogene in the initiation and progression of tumors. It has also been found as a potential marker in cancer detection and diagnosis. Moreover, it has been postulated to be useful in therapy signifying chemosensitivity and drug resistance [4].

MiR-675 is a derivative of lncRNA H19, existing as two variants miR-675-5p and miR-675-3p [5]. It was originally demonstrated to restrict the growth of the placenta prior to birth [6]. It has been strongly associated with BC and has been correlated with the histological grade of the cancer. It has been found to be implicated in cell viability, invasion and metastasis which has made it a potential target in breast cancer treatment [7]. MiR-let 7 was one of the first miRNAs discovered in *Caenorhabditis elegans*. The downregulation of let-7 has been linked with chemoresistance and the inhibition of growth in BC cells [3,8].

Several oncogenic pathways are involved in BC invasion and progression. Therefore, the use of functional assays can serve as potential novel markers in tumor progression. Currently used diagnostic biomarkers, such as CA 15-3 and CEA, although being widely used in cancer diagnosis, have been found to have a low positive rate, making the discovery of alternative diagnostic biomarkers with higher sensitivity and specificity a research pursuit [9]. The identification of valid conclusive biomarkers in the detection of BC and other types of cancers could replace the invasiveness of unnecessary pathology biopsies and the downsides of mammography.

## 2. Materials and Methods

### 2.1. Sample Collection

After approval of the Ethical Committee of the Medical Research Institute, Alexandria University, ninety female subjects were included in this study. Sample collection was carried out from October 2019 to January 2021. All patients were recruited from the Surgery and Oncology departments. Patients with diabetes mellitus, coronary artery disease, liver disease, asthma, sickle cell anemia, thyroid disease, as well pregnant women were excluded.

For all the subjects in the study, a full clinical evaluation and pathological investigation were recorded.

### 2.2. Sample Preparation

Five milliliters of blood were collected, 3 mL of which were placed in ethylene diamine tetraacetic acid (EDTA) tubes, then centrifuged for 10 min at 6000 rpm to collect plasma, and 2 mL were left to coagulate for 30 min then centrifuged for 10 min at 3000 rpm to collect the serum. The collected plasma was used for the complete blood count (CBC) analysis and for the assay of the ncRNAs in question. The serum was divided into two aliquots, one for the CA 15-3 analysis and the second for the routine biochemical tests. The routine biochemical tests and CA 15-3 were analyzed on the same day. The plasma samples were encrypted and stored at −80 °C till the time of analysis.

### 2.3. Laboratory Investigation

Complete blood picture and morphological examination, serum glucose, creatinine, alkaline phosphatase (ALP) activity, alanine aminotransferase (ALT) activity, aspartate aminotransferase (AST) activity, and cancer antigen 15-3 (CA 15-3)

### 2.4. RNA Extraction

Total RNA was isolated from the plasma samples using miRNeasy Mini Kit (Qiagen, Germany) according to the manufacturer instructions. The miRNeasy Mini Kit combines the phenol/guanidine-based lysis of samples and silicon membrane-based purification of total RNA. QIAzol Lysis Reagent, included in the kit, is a monophasic solution of phenol and guanidine thiocyanate, designed to facilitate the lysis of tissues, to inhibit RNases, and to remove most of the cellular DNA and proteins from the lysate by organic extraction.

### 2.5. Reverse Transcription

Reverse transcription was conducted using miScript II RT Kit (Qiagen, Germany) according to the manufacturer instructions. The miScript II RT Kit was used to perform a one-step, single-tube reverse transcription reaction. miScript HiFlex Buffer promotes the conversion of all RNA species (mature miRNA, precursor miRNA, noncoding RNA, and mRNA) into cDNA. This enables flexibility to study miRNA biogenesis and genome wide miRNA and mRNA regulation in a single cDNA sample.

### 2.6. Relative Quantification

Quantitative PCR was applied to determine the circulating levels of lncRNA H19, miR-675-5p, miR-675-3p and miR-let 7. It enables the real-time PCR quantification of mature miRNA and ncRNA using target-specific miScript Primer Assays (forward primers) and the miScript SYBR Green PCR Kit, which contains the miScript Universal Primer (reverse primer) and QuantiTect SYBR Green PCR Master Mix. The relative quantification (RQ) using comparative threshold cycle (Ct) provides an accurate comparison between the initial levels of template in each sample. The key feature of real-time PCR is that the amplified DNA is detected as the reaction progresses in real time and data is collected throughout the PCR process. A normalizer or reference gene (miRNA-16-5p) was used as an internal control for experimental variability in this type of quantification. This method of relative quantification is called the Livak method. Quantitative PCR assay was carried out using a Rotor-Gene SYBR Green PCR Kit (Qiagen^®^, Germany).

## 3. Results

### 3.1. Statistical Analysis

CA 15-3 serum levels were significantly higher in the two groups of BC patients who underwent surgery only or adjuvant chemotherapy (both pre and post treatment) compared with the control group. CA 15-3 dropped significantly after surgery yet did not show a significant change after chemotherapy (Figure 1).

The circulating H19 levels were significantly higher in the BC patients of the surgery group (pre and post-surgery) compared with the control group. The post-surgery level showed a significant decline compared with the presurgery level. Regarding the patients of chemotherapy group, the prechemotherapy levels of H19 are comparable with the control levels and significantly lower than the post-surgery level. The post-chemotherapy levels of circulating H19 are significantly lower than the control group (Figure 2).

The circulating miR-675-5p levels were significantly higher in the BC patients pre-and post-surgery compared with the control group. The post-surgery level showed significant decline compared with the presurgery level. Regarding the patients of the chemotherapy group, the prechemotherapy levels of miR-675-5p are not differ from the control levels, however, significantly lower than the post-surgery levels. The post-chemotherapy levels of circulating miR-675-5p are significantly higher than the control group (Figure 3).

The circulating levels of miR-675-3p showed significantly higher levels in all the BC patients; pre and post-surgery and pre and post-chemotherapy, compared with the control group. The highest circulating levels of miR-675-3p observed in the presurgery patients and the lowest levels were detected in the post-chemotherapy patients. The post-surgery levels were significantly lower than the presurgery. Moreover, the prechemotherapy levels were significantly lower than the post-surgery levels (Figure 4).

The circulating miR-let 7 levels were significantly higher in the presurgery and post-chemotherapy BC patients when compared to the control group. In the surgery group of patients, the post-surgery levels of miR-let 7 were significantly lower when compared with the presurgery levels. In contrast, in the chemotherapy group of patients, the post-chemotherapy levels of miR-let 7 were significantly higher than the prechemotherapy levels (Figure 5).

The Index (let 7/H19) combines the changes of two circulating markers, the alleged tumor suppressor microR-let 7 and the oncogenic lncRNA H19. The ratio was significantly lower in the presurgery BC patients compared to the control group. The ratio then showed a subsequent stepwise increase in the post-surgery, prechemotherapy, and post-chemotherapy patients. The highest ratio was observed in the post-chemotherapy BC patients, showing an almost seven-fold increase when compared to the control group. The ratio was significantly increased in the prechemotherapy group when compared with the post-surgery group. The level significantly increased in the post-chemotherapy group when compared with the prechemotherapy group (Figure 6).

### 3.2. Correlation Studies

A moderate inverse correlation was found between the circulating miR-let 7 levels versus CA 15-3 in the presurgery study group (Figure 7).

There is a direct strong correlation between H19 and total miR-675-5p levels (Figure 8) and a direct moderate correlation between H19 and miR-675-3p (Figure 9).

### 3.3. Relation Studies

Evaluation of the association of clinicopathological parameters in relation to the circulating ncRNAs measured in the present study showed a significantly increased level of miR-675-3p in patients with a positive family history versus those with no family history of BC. (Figure 10) Moreover, the circulating level of miR-let 7 was significantly associated with tumor size (Figure 11); a significantly lower level in tumor sizes of more than 2 cm when compared to tumors of size less than 2 cm.

No statistically significant association was found between the clinicopathological parameters and the circulating levels of ncRNAs in the prechemotherapy BC patients. (Figure 12, Figure 13 and Figure 14).

### 3.4. Validity Studies

Validity analysis studies to discriminate presurgery patients from control revealed the highest AUC for miR-675-3p, with an AUC of 1.00 (95%: 0.957—1.012) CI, a cut off value of 2.53, having a 100% sensitivity, specificity, positive and negative predictive value (PPV, NPV). CA 15-3 has AUC of 0.984 with a cut of value >13.1 U/mL, with a sensitivity of 96.7% and specificity of 93.3%. LncRNA H19 had an AUC of 0.956 with a cutoff value of > 1.65. MiR-675-5p has an AUC of 0.951 with a cut off value of >1.45, with a sensitivity of 90% and specificity of 83.3%. (Figure 15) Let 7/H19 ratio showed an AUC of 0.749 with a cut off value of ≤0.51, and sensitivity of 83.3% and specificity of 80%. A combination of CA 15-3 with lncRNA H19 yielded an AUC of 1.00 with sensitivity, specificity, PPV, and NPV of 100% (Figure 16).

## 4. Discussion

NcRNAs have been implicated in various hallmarks of cancer. Extensive research on ncRNAs has led to the understanding that lncRNAs as well as miRNAs are able to govern the levels of expression of their corresponding target genes at transcriptomic and translational levels. Gulìa et al. illustrated the effect of ncRNAs on the control of the metastatic, migration, and invasion processes as well as the effect on cell cycle arrest and proliferation of bladder cancer cells [10]. Moreover, novel treatment pathways are starting to emerge in the clinical practice using targeted inhibition of various ncRNA molecules. This would be of particular importance to women in their fertile years desiring to become pregnant. This is because fertility rates have been found to be about three times lower in women with a history of BC treatment, due to the need to delay conception until treatment is complete and the direct gonadotoxic effects of treatments [11]. A futuristic yet legitimate outlook to the power of assessing ncRNAs can combine diagnostic, prognostic, and therapeutic parameters to help in generating an ‘epigenetic profile’ of cancer patients and to derive a tailored therapy for cancer patients of different stages and subtypes [12].

In the present study, CA 15-3 was measured as a conventional circulating tumor marker in breast cancer for comparison of the ncRNAs under investigation. CA 15-3 serum levels were significantly higher in the two groups of BC patients who underwent surgery only and who received adjuvant chemotherapy, compared with the control group. CA 15-3 dropped significantly after surgery yet did not show a significant change after chemotherapy.

H19 is involved in many biological processes including apoptosis, cell proliferation, and invasion in several tumors including BC [13]. It is implicated in the development of BC and increases G1 to S cell cycle transition when up regulated [14]. High tumor expressions of H19 were associated with the size of the tumor, nodal status and hormone negativity, having a negative correlation with the prognosis of BC. Unfavorable disease free and overall survival were exhibited in patients with high H19 levels [15]. Moreover, another study has found that the overexpression of H19 brought about a reversal in the tumor inhibitory effects of Huaier extract, yet its knocking down sensitized BC cells to Huaier extract. This study promoted the idea that the H19/miRNA-675/CBL pathway is responsible for the BC cell proliferation and induction of apoptosis [7]. In light of these findings, H19 serves as a potential biomarker for the diagnosis of BC as well as a useful tool in the assessment of the response to adjuvant and neoadjuvant chemotherapy in BC. Furthermore, the response to therapy was also correlated to H19, where overly expressed H19 increased BC cell drug resistance to doxorubicin [16]. The findings of the present study agree with the emerging evidence identifying H19 as an oncogene. Analysis of the circulating H19 in the present study revealed significantly higher levels in the presurgery BC patients when compared to the control group. The levels significantly dropped after surgery yet were still significantly higher than the control group. A further significant drop in the BC patients about to receive adjuvant chemotherapy was measured with no significant change in circulating H19 upon receiving chemotherapy. Then, the circulating H19 level showed a further significant drop in the prechemotherapy group when compared to the postsurgery BC patients and the chemotherapy did not show further significant change in H19 levels.

With regard to the descendants of H19; miRNA 675-5p and 675-3p, a literature review found that miRNA-675, whether alone or together with H19, plays a role in cellular invasion and migration [17]. A study of miRNA-675 in paraffin embedded tissues demonstrated that the expression of miRNA-675 was significantly higher in the tissues of patients with BC compared with the control group, where no association was found between expression levels and age, lymph node, stage and receptor status [18]. In agreement with this data, Matouk et al. demonstrated that miRNA-675 indirectly targets slug leading to increase of cell invasion and in vivo metastasis [19]. Another study has also identified both H19 and miRNA-675 as key players in the activation of the invasion and migration of BC cells [17]. These findings contrast with the results of a study carried out on 63 non-metastasized BC patients in University Medical Center Hamburg-Eppendorf, where no deregulation of plasma miRNA-675 was found in the entire BC group [18]. The discrepancy in the results of miRNA-675 in BC patients may be reasoned by the method of analysis and calculation of data. As there is no consensus concerning the data normalization in the circulation, miRNA-16-5p was chosen as a reference gene in the present study to normalize the lncRNA and microRNAs’ data since miR-16-5p is one of the most frequently used reference gene according to literature [20]. Meanwhile, in the study of Müller et al. [18], miR-484 and b-actin were used as references to normalize their miRNA and lncRNA data, respectively.

Findings of the present study showed an increase in the levels of miRNA descendants of H19; miR-675-5p and miR-675-3p, in the pre- and post-surgery BC patients when compared to the control group. The post-surgery levels of miR-675-5p showed a significant decline compared with the presurgery level. In the BC patients receiving adjuvant chemotherapy, the prechemotherapy levels of miR-675-5p are not statistically different from the control levels, however significantly lower than the post-surgery group’s levels. The circulating levels of miR-675-3p (passenger strand of miR-675-5p) showed more prominent increase in BC patients compared to miR-675-5p. The highest circulating levels of miR-675-3p were observed in the presurgery patients while the lowest levels were detected in the post-chemotherapy patients. The post-surgery levels were significantly lower than the presurgery. Moreover, the prechemotherapy levels were significantly lower than the post-surgery levels.

Literature regarding miR-let 7 in BC has been controversial. On one hand, let-7a is well characterized as a tumor suppressor, with downregulated levels in several solid organ cancers [21]. It has also been linked to the chemosensitivity of cancer in vitro cell lines, where miR-let 7 has been found to be an important predictor for the clinical response to the anthracycline drug epirubicin. Hence, it has been suggested to be a therapeutic target to regulate resistance from epirubicin-based chemotherapy [22]. Research findings indicate that lower expression of let-7a miRNA can induce chemoresistance in breast cancer by enhancing cellular apoptosis and suggest that let-7a may be used as a therapeutic target to modulate epirubicin-based chemotherapy resistance [22]. This phenomenon, although proven in cell lines, has not yet been proven in clinical breast tumors. On the other hand, miR-let 7 levels were unexpectedly almost 11.2-fold increased in BC patients when compared with the control group. After tumor resection, levels significantly dropped to levels comparable to that of the control group [23]. These findings were in line with the results of the present study which show significantly increased levels of circulating miR-let 7 in the presurgery and post-chemotherapy BC patients when compared to the control group. Although initially high in the BC patients, surgery brought upon a significant reduction. In contrast, chemotherapy brought about a significant increase in the levels of miR-let 7 in the BC group receiving adjuvant chemotherapy.

The finding that miR-let 7 was greatly increased in the blood of BC patients raises an interesting question concerning the origin of the circulating miRNAs. The mechanism of delivery of tumor associated ncRNAs into the bloodstream remains unknown. Two hypotheses were raised by Slack et al. in a report. Primarily, tumor cell lyses and death releases ncRNAs into the circulation. A second hypothesis is that tumor cells expel ncRNAs across new blood vessels and pour into the circulation of the microenvironment of the tumor [24]. The upregulation after chemotherapy could indicate that the patients in this study showed a hypothetically good response to chemotherapy and could possess a good prognosis for their treatment, yet the upregulation in the non-intervened BC patients could be a feedback mechanism for the cells in response to the elevated H19 levels.

From the assumption above, a consequent ratio was derived from the present work of miR-let 7 to H19 demonstrating a theoretical tumor suppressor to oncogene level analysis. Statistical analysis of this ratio revealed that the presurgery group had a significantly lower ratio when compared with the control group. This ratio increased significantly in the prechemotherapy group when compared to the post-surgery group and showed a further statistically significant increased after receiving chemotherapy, also becoming significantly higher than the control group in the post-chemotherapy group.

The results of the present study, in terms of the effect of surgery on the levels of the ncRNAs, revealed a significant decrease in the levels of CA 15-3, H19, miR-675-5p, miR-675-3p, and miR-let 7. The ratio of miR-let 7 to H19 showed no statistical difference after surgery. Chemotherapy on the other hand, showed no statistically significant difference between the prechemotherapy and post-chemotherapy levels of H19, miR-675-5p and miR-675-3p. MiR-let 7 levels as well as let 7 to H19 ratio showed a statistically significant rise in the post-chemotherapy group when compared with the prechemotherapy group.

Correlation studies from the present work identified a direct strong correlation between H19 and miR-675-5p (r = 0.753, *p* < 0.001) and a direct moderate correlation between H19 and miR-675-3p (r = 0.448, *p* < 0.05). This could be explained as an increase in precursor H19 results in a consequent increase in its descendants miR-675-5p and -3p. Although this finding agrees with most of the studies assessing the H19/ miR-675 in BC patients, a striking finding was found where opposite levels were reported with an increase in H19 accompanied by a decrease in miRNA-675 in the HER2-positive BC subgroup [15]. No correlation was found between H19 and let7 levels in the BC in the present study, yet an inverse moderate correlation between CA 15-3 and miR-let 7 level was found (r = −0.0364, *p* < 0.05). This goes in alignment with literature identifying CA 15-3 as a tumor invasion promoter and miR-let 7 as a tumor suppressor.

Relation studies were evaluated in the presurgery group assessing the effect of family history, menopause, grade, lymph invasion, tumor size, and stage. No relation was found in the levels of the ncRNAs in the present study with regards to the menopausal state, grade and stage. No relation was also found in nodal status subgroups, in agreement with a previous study performed on BC tissue of patients finding no differences between nodal status subgroups [15]. The results revealed a significantly increased miR-675-3p level in patients with a positive family history of breast cancer (*p* < 0.001). From this finding, we could derive a non-invasive method for the surveillance of circulating miR-675-3p in individuals with a positive family history of BC. An inverse relation was also identified from the current study between tumor size and miR-let 7, where a significantly lower level in tumor sizes of more than 2 cm as compared with tumors of less than 2cm (*p* < 0.001). This could potentially provide a more accurate estimate of tumor size before surgery, being of substantial value in neoadjuvant therapy, where tumor size could be monitored easily and directly using circulating miR-let 7 levels.

Among the chemotherapy group, relation studies relating receptor status, ER, PR, and HER2 were analyzed together with their intrinsic subtyping classification and the triple negativity. Although no statistically significant results were concluded, miR-675-3p levels were notably higher in the HER2 positive group when compared with the negative group with a significance of 0.077. This shows similar findings to the study by Müller et al. [15], where an association was found between circulating H19 and HER2 overexpression [15]. They were also remarkably higher in the luminal B group when compared to the luminal A group. These difference although have not proven significance in the present study, indicate that levels of miR-675-3p are higher in subgroups with worse prognosis. More conclusive results could be attained by performing the study on a larger sample size of BC patients.

Validity analysis of the parameters measured in this study displayed a maximal AUC for miR-675-3p at a cut off value of >2.53, with an AUC of 1.000 and 100% sensitivity, specificity, PPV, and NPV. CA 15-3 produced an AUC of 0.984 (95% CI: 0.957—1.012) with a cut of value >13.1 U/mL, having a sensitivity and specificity of 96.7% and 93.3% respectively. Although exhibiting an excellent sensitivity and specificity in this study, a cut off value of 13.1 U/mL is not a reflection of the acceptable reference level used by the laboratory standards. Analysis of H19 shortly followed with an AUC of 0.956 (95% CI: 0.895—1.016) with a cutoff value of >1.65 and a sensitivity and specificity of 90.0 and 86.67 respectively, making it a very strong diagnostic tool for BC. MiR-675-5p produced an AUC of 0.951 (95% CI 0.893—1.009) with a cut off value of >1.45, a sensitivity of 90%, and specificity of 83.3%. Let 7 exhibited an AUC of 0.941 (95% CI: 0.887—0.995) at a cut off value > 1.32 with a sensitivity and specificity of 96.67% and 80%, respectively. The let 7/ H19 index yielded a weaker diagnostic power with an AUC of 0.749 (95% CI: 0.613—0.885) with a cut off value of ≤0.51, sensitivity of 83.3% and specificity of 80%. The combination of the conventional CA 15-3 with the H19 in the present study further enhanced the discriminative power of this test, producing a viable complement to current BC detection strategies, yielding an AUC of 1.000 with 100% sensitivity, specificity, PPV, and NPV. In conclusion, the entire panel of ncRNAs in this study can potentially act as novel biomarkers whether alone or combined in the diagnosis of BC. Surgery results in a significant drop in the levels of all H19, miR-675-5p, miR-675-3p, and miR-let 7. Adjuvant chemotherapy did not result in a profound effect on the levels of ncRNAs in this study, yet it resulted in a statistically significant increase in the level of miR-let 7. Significantly increased miR-675-3p was associated with a positive family history of BC, making it a potential tool for assessing the onset of disease in females with a family history of BC. Elevated miR-let 7 levels were associated with smaller tumor sizes and, if validated, could provide a non-invasive method for tumor size surveillance.

The present study was not free from limitations. Results could be further validated and verified by adopting a larger sample size to obtain a more representative significance that is true to value, measuring the levels of ncRNAs in another group of subjects with benign tumors, following through with the same non-intervened BC patients, collecting blood before and after surgery, as well as before and after chemotherapy for the same patients, and assessing the levels of the studied panel of ncRNAs in patients receiving neoadjuvant chemotherapy followed by surgery.

## 5. Conclusions

The circulating levels of ncRNA measured in the present study could serve as a gold mine in breast cancer pathogenesis and treatment, providing valuable solutions in the clinical setting considered.

## Figures and Tables

**Figure 1 biomedicines-11-00563-f001:**
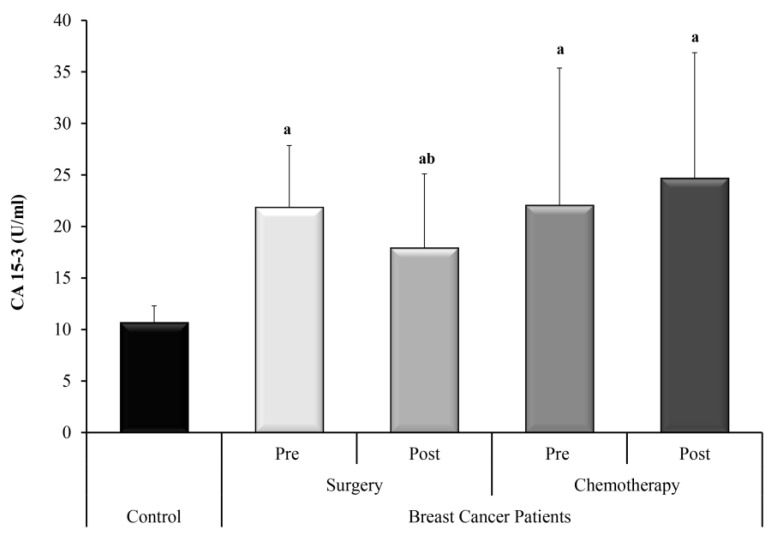
The levels of CA 15-3 in control subjects and BC groups; surgery and adjuvant chemotherapy. a—Significant between control and each group by using Post Hoc Test (Tukey) for ANOVA test, b—Significant between pre and post in same group by using Paired *t*-test.

**Figure 2 biomedicines-11-00563-f002:**
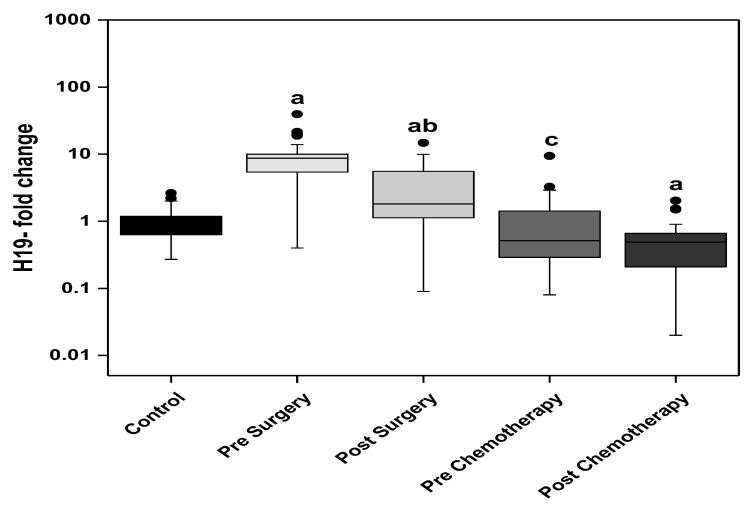
The circulating levels of H19 in the control subjects and the BC studied groups; surgery only and adjuvant chemotherapy. Extreme value: a—Significant between control and each group by using Post Hoc Test (Dunn’s), Kruskal Wallis test, b—Significant between pre and post in same group by using Wilcoxon signed ranks test, c—Significant between post-surgery and prechemotherapy by using Mann Whitney test.

**Figure 3 biomedicines-11-00563-f003:**
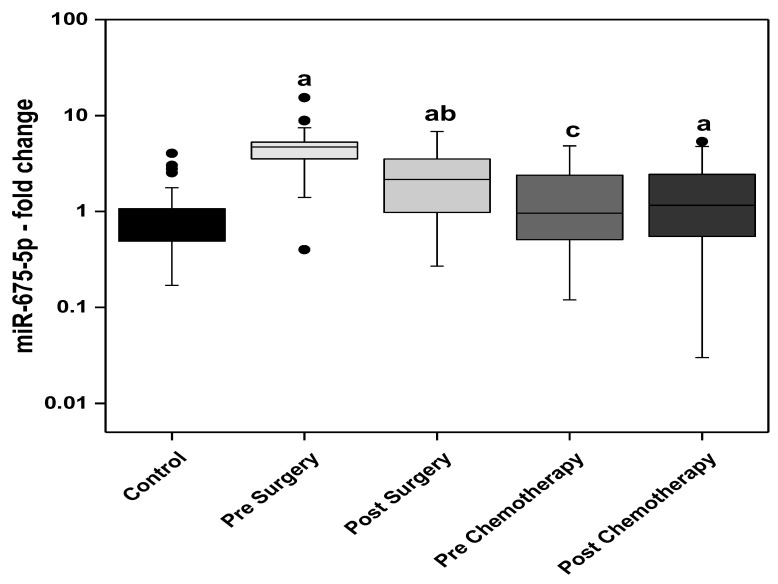
The circulating levels of miR-675-5p in the control subjects and the BC studied groups; surgery only and adjuvant chemotherapy. Extreme value: a—Significant between control and each group by using Post Hoc Test (Dunn’s), Kruskal Wallis test, b—Significant between pre and post in same group by using Wilcoxon signed ranks test, c—Significant between post-surgery and prechemotherapy by using Mann Whitney test.

**Figure 4 biomedicines-11-00563-f004:**
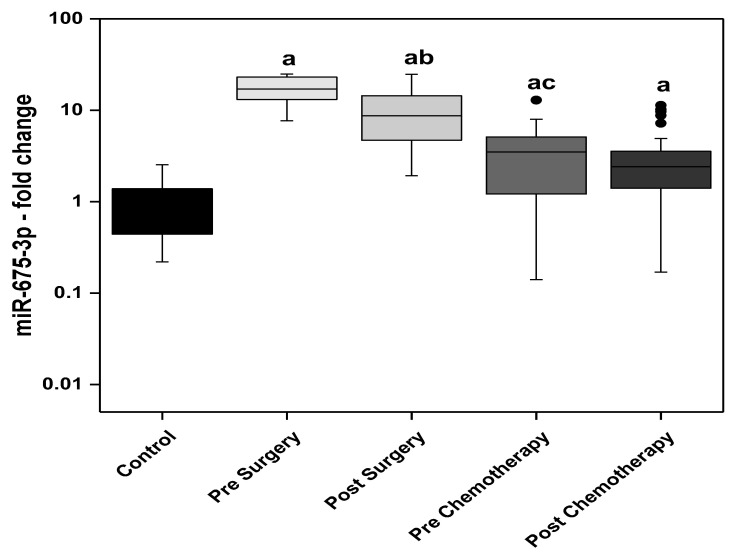
The circulating levels of miR-675-3p in the control subjects and the BC studied groups; surgery only and adjuvant chemotherapy. Extreme value: a—Significant between control and each group by using Post Hoc Test (Dunn’s), Kruskal Wallis test, b—Significant between pre and post in same group by using Wilcoxon signed ranks test, c—Significant between post-surgery and prechemotherapy by using Mann Whitney test.

**Figure 5 biomedicines-11-00563-f005:**
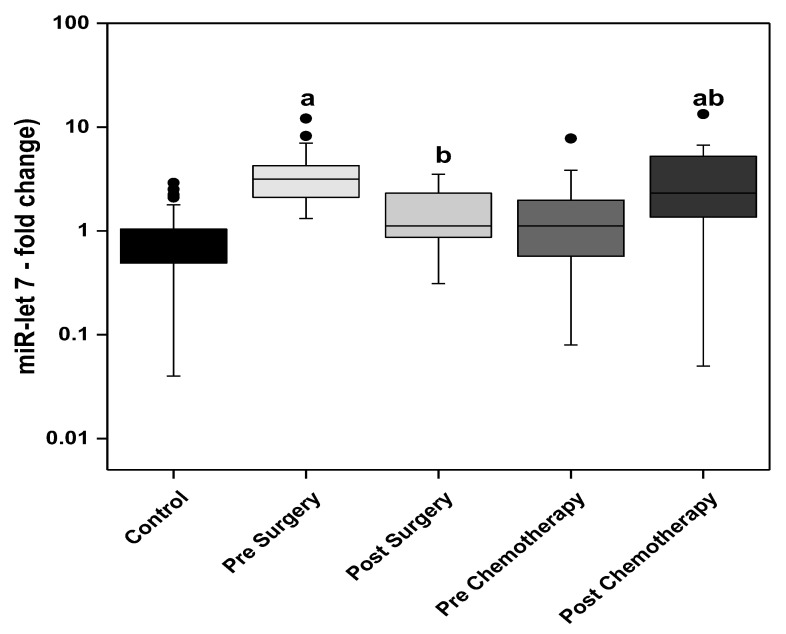
The circulating levels of miR-let 7 in the control subjects and the BC studied groups; surgery only and adjuvant chemotherapy. Extreme value: a—Significant between control and each group by using Post Hoc Test (Dunn’s), Kruskal Wallis test, b—Significant between pre and post in same group by using Wilcoxon signed ranks test.

**Figure 6 biomedicines-11-00563-f006:**
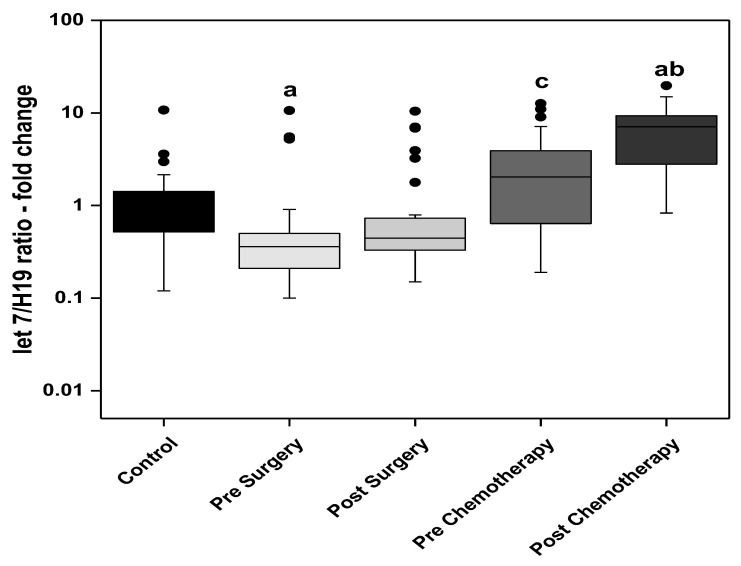
The let 7/H19 ratio in the control subjects and the BC studied groups; surgery only and adjuvant chemotherapy. Extreme value: a—Significant between control and each group by using Post Hoc Test (Dunn’s), Kruskal Wallis test, b—Significant between pre and post in same group by using Wilcoxon signed ranks test, c—Significant between post-surgery and prechemotherapy by using Mann Whitney test.

**Figure 7 biomedicines-11-00563-f007:**
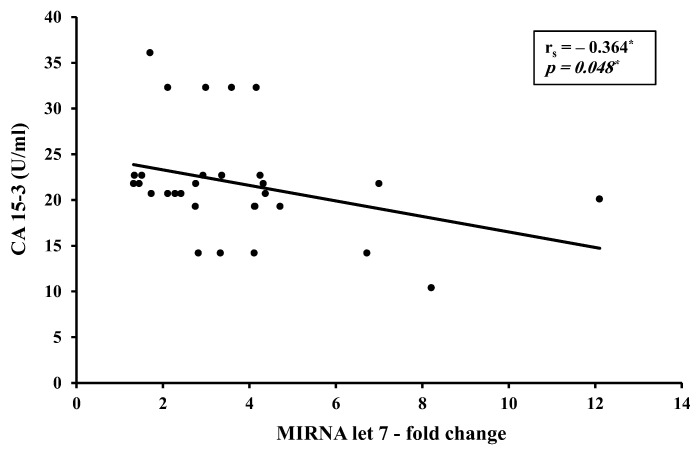
Correlation curve between CA 15-3 and circulating miR-let 7 (*n* = 30). * Statistically significant at *p* ≤ 0.05.

**Figure 8 biomedicines-11-00563-f008:**
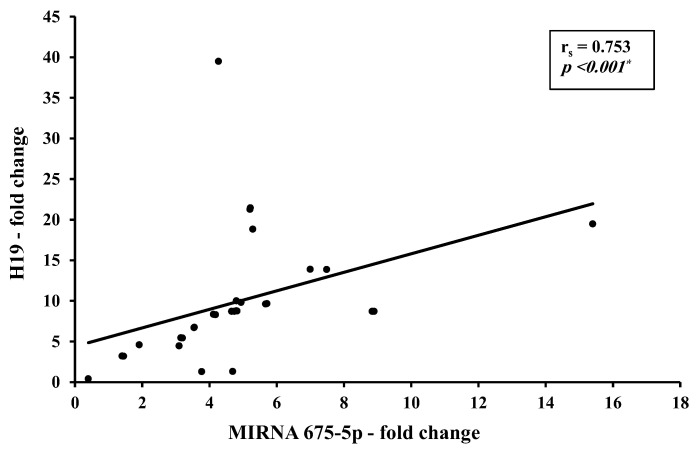
Correlation between circulating H19 with miR-675-5p in presurgery BC patients (*n* = 30). * Statistically significant at *p* ≤ 0.001.

**Figure 9 biomedicines-11-00563-f009:**
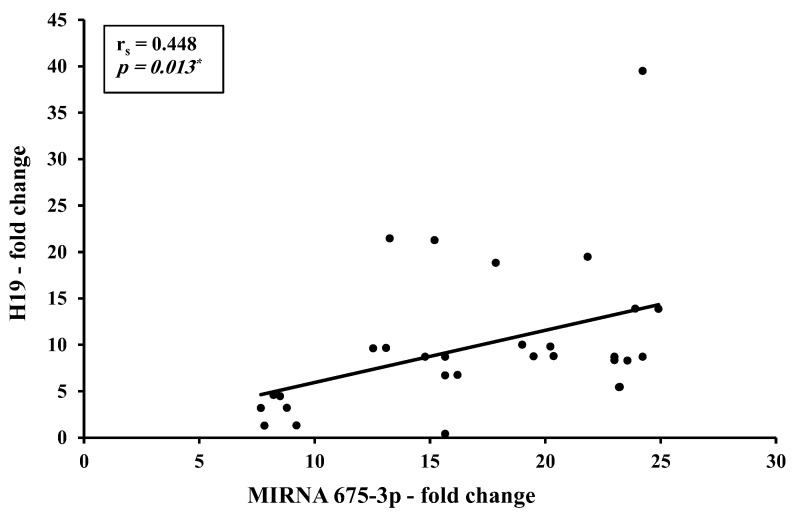
Correlation between circulating H19 with miR-675-3p in presurgery BC (*n* = 30). *: Statistically significant at *p* ≤ 0.05.

**Figure 10 biomedicines-11-00563-f010:**
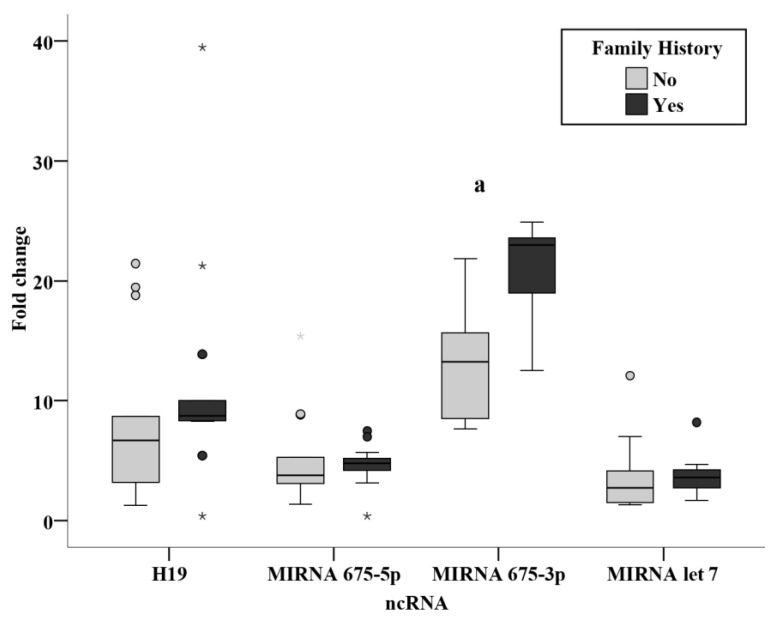
Relation between ncRNAs and family history in presurgery BC patients. *—Extreme value: a—Significant between ncRNA and FH using Mann Whitney test.

**Figure 11 biomedicines-11-00563-f011:**
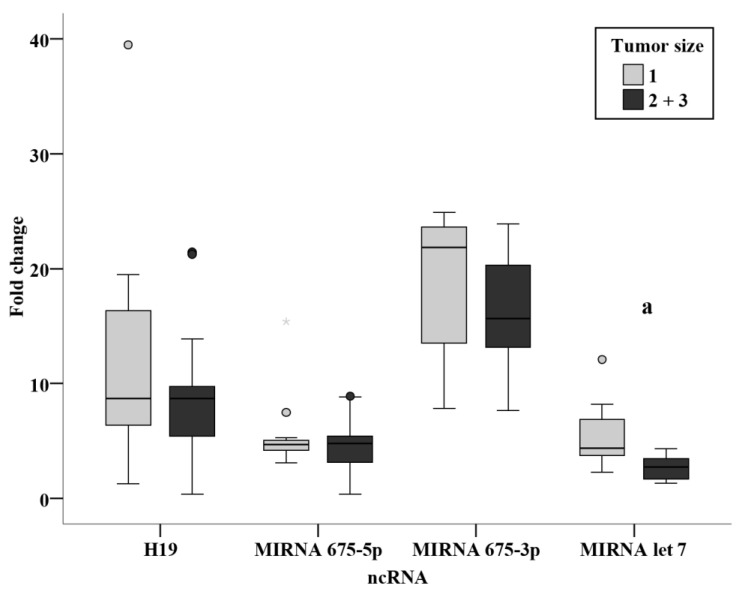
Relation between ncRNA and tumor size in presurgery BC patients, *—Extreme value: a—Significant between ncRNA and tumor size using Mann Whitney test.

**Figure 12 biomedicines-11-00563-f012:**
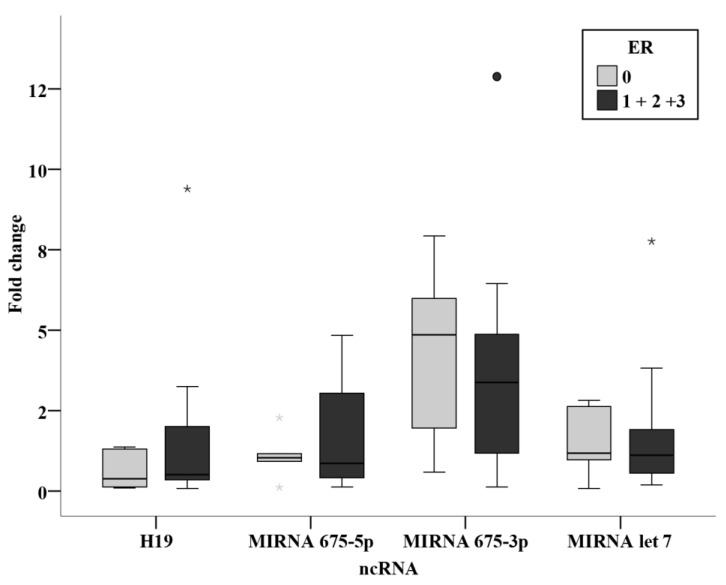
Relation between ncRNAs and ER status in the prechemotherapy BC patients. *—Extreme value.

**Figure 13 biomedicines-11-00563-f013:**
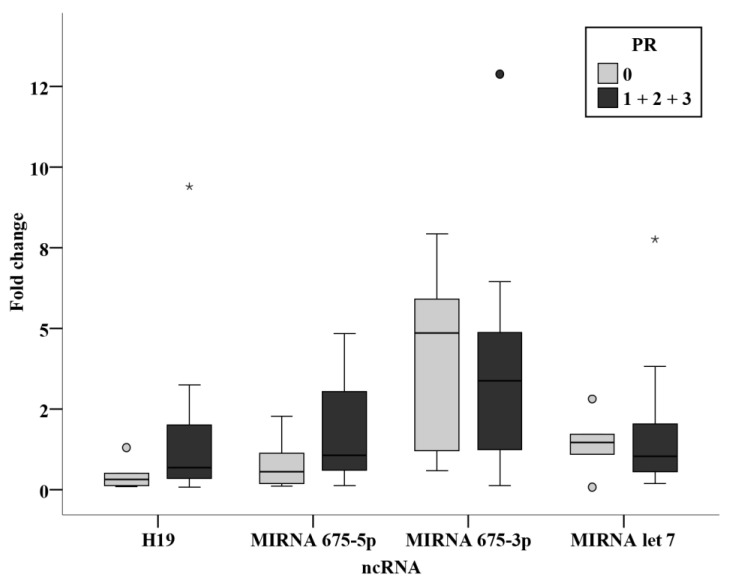
Relation between ncRNAs and PR status in the prechemotherapy BC patients. *—Extreme value.

**Figure 14 biomedicines-11-00563-f014:**
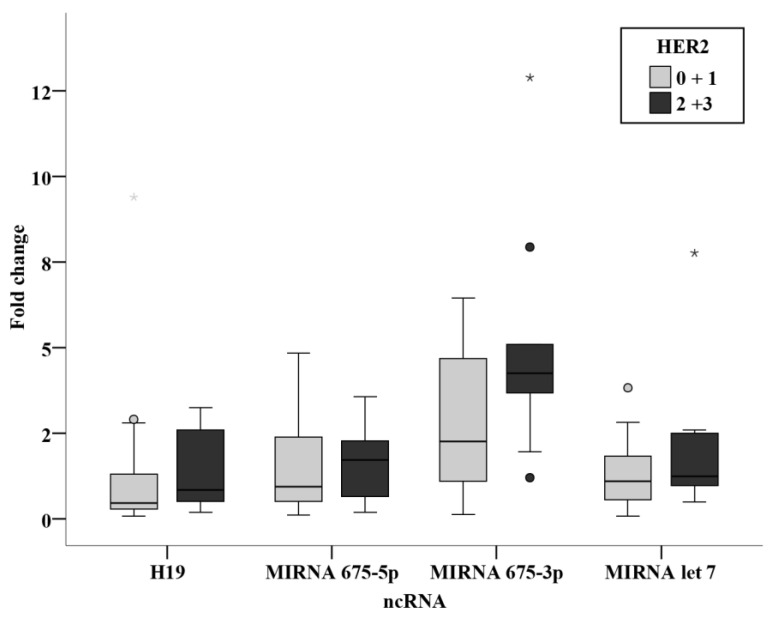
Relation between ncRNAs and HER2 status in the prechemotherapy BC patients. *—Extreme value.

**Figure 15 biomedicines-11-00563-f015:**
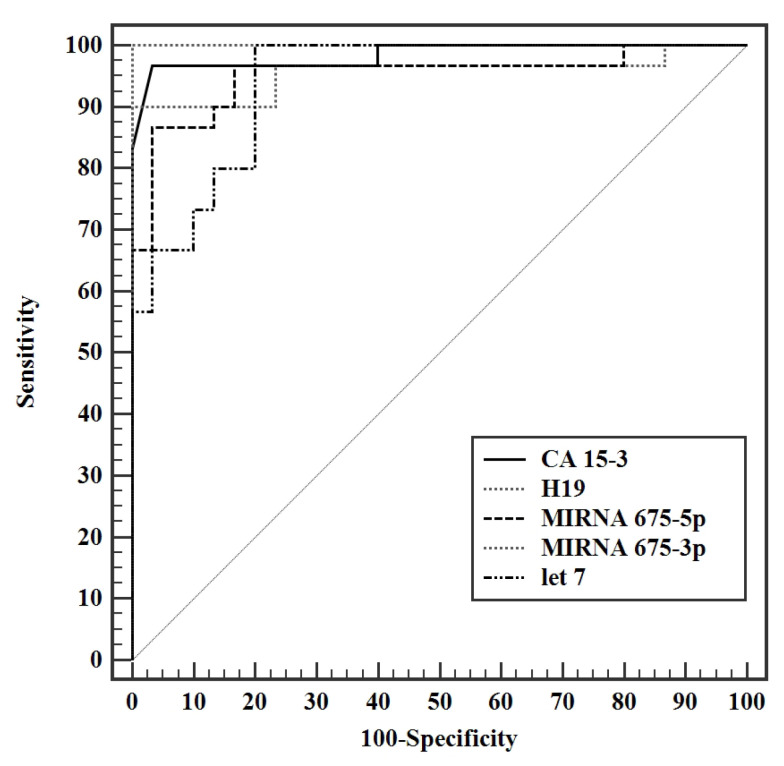
ROC curve for CA 15-3 (U/mL), H19, MIRNA 675-5p, MIRNA 675-3p and let 7 to discriminate pre surgery patients (n = 30) from control (n = 30).

**Figure 16 biomedicines-11-00563-f016:**
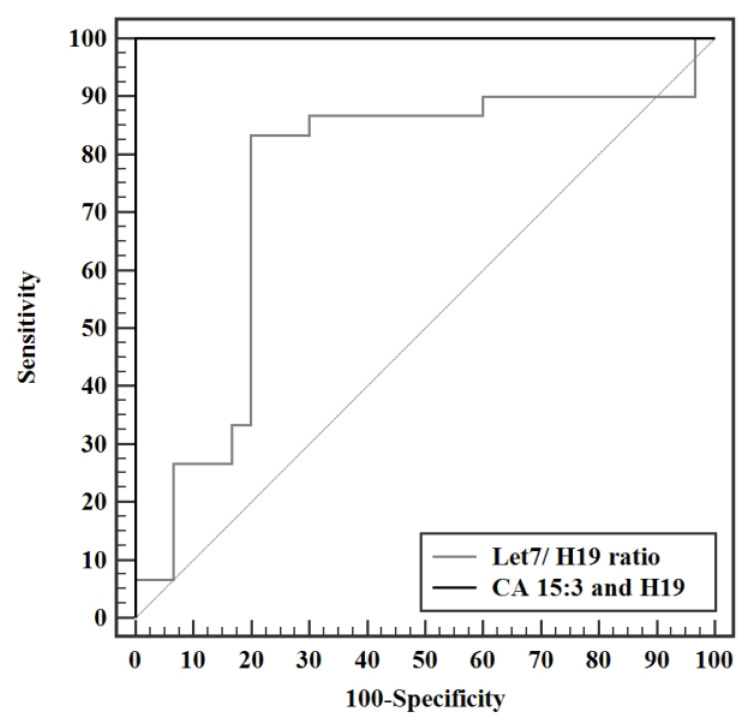
ROC curve for let7/H19 ratio and combined CA 15:3 and H19 to discriminate pre surgery patients (n = 30) from control (n = 30).

## Data Availability

Data supporting reported results can be provided upon request.

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
