# Peer review of "A Panel of Circulating Non-Coding RNAs in the Diagnosis and Monitoring of Therapy in Egyptian Patients with Breast Cancer"

_biomedicines, 2023, doi:10.3390/biomedicines11020563_

Round 1

Reviewer 1 Report

This case-control study aimed to investigate the relative quantification of long non-coding RNA, microRNA and miR-let in breast cancer patients and found out that panels of ncRNAs in this study can all potentially act as novel biomarkers whether alone or combined in the diagnosis of breast cancer.  The topics is of interest and i would like to congratulate with Authors for their effort:

My comments are:

1)   Add in the discussion section a word on the importance of using specific lncRNA as biomarkers or therapeutic targets [Gulìa C., Baldassarra S., Signore F. et al Role of Non-Coding RNAs in the Etiology of Bladder Cancer. Genes 2017, 8, 339; doi:10.3390/genes8110339] and on personalized medicine with  tailor treatments to patients thanks to nc-rNA . [Cavaliere AF., Perelli F., Zaami S., Piergentili R. et al. Towards personalized medicine: Non-coding rnas and endometrial cancer. 2021. Healthcare (Switzerland); 9. DOI: 10.3390/healthcare9080965;

R PiergentiliS Zaami, AF Cavaliere et al. Non-coding rnas as prognostic markers for endometrial cancer. International Journal of molecular sciences. 2021, 22(6), 3151; https://doi.org/10.3390/ijms22063151]

2)   In the discussion section a word on the fact that the number of women in reproductive age with a history of BC is significantly increasing, and many BC survivors desire fertility and become pregnant. Fertility rate has been reported to be about three times lower than in woman without history of BC mostly due to the direct gonadotoxic effects of treatments or the need to delay conception until the end of the therapies. Beyond the risk of infertility, potential risk of adverse obstetrical outcomes has been shown among BC survivors. It is therefore important to find new treatment strategies. D’Ambrosio V, Vena F, Di Mascio D. et al. Obstetrical outcomes in women with history of breast cancer: a systematic review and meta-analysis. Breast Cancer Research and treatment.  2019;178(3):485-492.]

Author Response

Thank you for your valuable review. 

The references provided have been indeed added to the discussion section.

Reviewer 2 Report

The manuscript “A Panel of Circulating Non-coding RNAs in the Diagnosis and Monitoring of Therapy in Egyptian Patients with Breast Cancer” by Nadine Wehida et al. is well written and properly designed. This study aims to assess the usefulness of long non-coding RNA (lncRNA) H19, microRNA (miR) 675-5p, 675-3p and miR-let 7 in breast cancer patients. For this reason there should be a solid data on the BC group characteristics i.e. TNM. There is enormous variability in different stages of BC also in the context of miRNA and other ncRNAs content. One of the reasons is also EMT that gives one more reason for the variability of the profile of target particles. With no data on the patients in this area, any conclusions made will be very elusive. Although, even in that case, results may not be conclusive enough to claim that just a couple of RNA polymers could constitute universal diagnostic or prognostic factors.

Thus the manuscript, although very interesting, requires some further data analysis.

Author Response

The selection criteria included non-metastasized BC patients, hence stages 1, 2 and 3, to be able to qualify for surgery. The clinicopathological characteristics of the patients in this study including their TNM staging distributions have been recorded and the appropriate relation studies were carried out. Evaluation of the association of clinicopathological parameters in relation to the circulating ncRNAs measured in the present study showed a significantly increased level of miR-675-3p in patients with a positive family history versus those with no family history of BC. Also, the circulating level of miR-let 7 was significantly associated with tumor size; a significantly lower level in tumor sizes of more than 2 cm when compared to tumors of size less than 2 cm. Nonetheless, given the sample size of the patients and the distribution of the results, it was not credible to derive conclusions from their staging.

Round 2

Reviewer 2 Report

I found the manuscrupt complete after modifications.